# Penetration of Non-Adhesive Gel-like Embolic Materials During Dural Vessels Embolization According to Characteristics of Tantalum Powder

**DOI:** 10.3390/jfb15110319

**Published:** 2024-10-27

**Authors:** Andrey Petrov, Arkady Ivanov, Sergei Ermakov, Egor Kolomin, Anna Petrova, Oleg Belokon, Konstantin Samochernykh, Larisa Rozhchenko

**Affiliations:** 1Vascular Neurosurgery Department, Polenov Neurosurgical Research Institute, Branch of Almazov National Medical Research Centre, 191014 Saint Petersburg, Russia; egor96kolomin@gmail.com (E.K.); petrovaanna2803@gmail.com (A.P.); neurobaby12@gmail.com (K.S.); rozhch@mail.ru (L.R.); 2Stavropol Krai State Budgetary Healthcare Institution “Stavropol Krai Clinical Hospital”, Semashko St., 1, 355030 Stavropol, Russia; s.v.yermakov@yandex.ru (S.E.); belokonoleg@gmail.com (O.B.); 3Belostrov Clinic of High Technologies, Clinic Beloostrov, 1, Yukki Urban Settlement, Vsevolozhsk District, Leningrad Region, 188651 Saint Petersburg, Russia

**Keywords:** gel-like embolic agent, tantalum powder, ONYX, SQUID, chronic subdural hematoma (CSDH), dural vessels

## Abstract

Tantalum powder is included in the composition of Non-Adhesive Gel-like Embolic Materials (NAGLEMs) for X-ray opacity. The duration of X-ray opacity during embolization is primarily associated with the particle size, which differs in the most used NAGLEMs—ONYX (Medtronic) and SQUID (Balt). NAGLEMs are widely used for the embolization of branches of the middle meningeal artery (MMA) in patients with chronic subdural hematomas (CSDHs). Considering the size (5–15 microns) of the target dural vessels, we assumed that not only the viscosity of NAGLEMs, but also the size and shape of tantalum granules may be important for the penetration of these gel-like embolic agents and determine their behavior. A notable discrepancy in size was observed. The medium-sized granules in the SQUID 18 sample (0.443 ± 0.086 microns, M ± SD) were found to be approximately ten times smaller than the tantalum granules in the ONYX 18 sample (5.2 ± 0.33 microns, M ± SD).Tantalum granules in SQUID 18 have a regular spherical shape; in ONYX 18 they have an irregular angular shape. When comparing the behavior of gel-like embolic agents of the same viscosity during MMA embolization in patients with CSDHs (an average age of 62.2 ± 14.3 years) in the group where SQUID 18 (*n* = 8) was used, the gel-like embolic agent in dural vessels demonstrated significantly greater penetration ability compared with the group where ONYX 18 (*n* = 8) was used. Accordingly, not only the viscosity of NAGLEMs, but also the size and shape of tantalum granules can have a significant effect on the penetration ability of gel compositions.

## 1. Introduction

The selection of an embolic agent has a significant impact on the outcomes of endovascular interventions [1]. Over the past decade, this has been particularly relevant for the rapidly evolving technology of endovascular treatment for chronic subdural hematomas (CSDHs) [2,3,4,5,6]. Systematic literature reviews [7,8,9,10,11,12,13,14] and results from the last three randomized controlled trials (EMBOLISE, MAGIC-MT, and STEM) [15,16,17], which were focused on this condition, highlight the benefits and significance of using liquid embolic agents. Among liquid embolic agents, those based on solutions of biologically inactive polymers in biocompatible and water-soluble organic solvents—so-called non-adhesive gel-like embolization materials (NAGLEMs)—are of particular interest [1,18,19]. These gel-like embolic agents solidify within the vascular bed through precipitation; this process is often compared to that of lava solidifying [1]. ONYX (Medtronic) and SQUID (Balt) are the most commonly used and studied examples of NAGLEMs. Thus, in the two most recent randomized studies on CSDHs (EMBOLISE and Magic-MT), ONYX 18 was used and, in the third study, (STEM)SQUID 18. One of the reasons for choosing embolization for the treatment of chronic subdural hematomas (CSDHs) can be found in the characteristics of blood supply of the meninges and capsule. The main arteries of the meninges are clearly visible during angiography and lie on the outer surface of the dura mater. These vessels have a large inner diameter enough to accommodate all types of embolic material, approximately 400–800 μm in diameter [2,20].

The so-called “penetrating arterioles” provide a connection between the inner surface membrane and the membrane of the CSDH. Their diameter ranges from 5 to 15 μm, as determined by Kerber C et al. (1973) [20]. Additionally, there are arterio-venous shunts in the middle layer of the dura, measuring 8–12 μm in diameter, that may also contribute to CSDH formation. Therefore, in endovascular treatment for CSDHs, target vessel diameters range from 5 to 15 microns. The main characteristics of NAGLEMs for endovascular surgeons include radiopacity, penetrability, distribution, and controllability. The penetrability properties of NAGLEMs determine the ability to occlude target vessels from the blood supply during MMA embolization.

The depth of penetration of embolic agents may be contingent upon two factors: the embolization technique, including the pressure inside the catheter, volumetric rate of insertion, and duration of reflux [21,22,23], and the viscosity and rheology characteristics of the polymer composition [1].

Standard-viscosity gel-like agents (low, medium, and high) can be effectively used for the embolization of highly vascularized lesions, but they do not always permit deep penetration of the gel-like embolic agent into the microvasculature. Ultra-low-viscosity gel-like agents have special embolization properties that significantly differentiate them from standard agents [19]. It is believed that, due to their lower viscosity and increased fluidity, these agents may provide faster and deeper antegrade penetration of the gel-like embolic agent and, consequently, lead to more effective penetration of the embolic material into ultra-fine vessels [24,25].

The viscosity and rheological properties of NAGLEMs depend not only on the concentration of the active ingredient (ethylene vinyl alcohol (EVOH)) but also on a number of factors, including the specificity of the contrast agent used, which can have a pronounced effect on the sedimentation of polymers in the vascular network [22,26].

Tantalum (Ta) is a dense, ductile, refractory metal whose powder is an effective radiopaque contrast agent in NAGLEMs where specific thermal properties, chemical resistance or biocompatibility, and high radiopacity are required [27,28].

As an illustration, the attributes of tantalum powder that influence the functionality of the most prevalent NAGLEMs (ONYX and SQUID) are well documented. Although this statement remains controversial [19,29], it is believed that the micronized tantalum powder contained in all SQUID variants results in fewer beam enhancement artifacts on CT and flat panel imaging than with ONYX 18 [30]. However, during prolonged embolization procedures with ONYX 18, tantalum particles begin to sediment in the syringe and microcatheter after only a few minutes, which may reduce the visibility of ONYX or lead to microcatheter occlusion [19]. Basic information about the characteristics of the most popular NAGLEMs according to the literature is given in Table 1.

Therefore, the majority of studies examining tantalum particles in NAGLEMs concentrate on the sedimentation rate and, consequently, the duration of X-ray opacity. However, there is a lack of consideration for the potential influence of powder properties on penetration ability. It is plausible that the behavior of NAGLEMs may be influenced by factors such as particle size distribution, granule shape, and powder purity when penetration into a small diameter vasculature is required.

Given the size of target dural vessels (5–15 microns) during MMA embolization of patients with CSDHs, it is possible that not only the viscosity, but also the size and shape of the tantalum particles contained in the NAGLEM product may play a role in the penetration of these gel-like embolic agents and determine its behavior.

## 2. Materials and Methods

### 2.1. Materials

To analyze the granules of tantalum powder, which are part of the most popular NAGLEMs, two samples were selected from a batch used in standard clinical practice.

Sample 1—gel-like composition ONYX 18 1.5 mL

Sample 2—gel-like composition SQUID 18 1.5 mL

The two compositions in question both contain the same polymer, namely an ethylene-vinyl alcohol copolymer with the abbreviated chemical formula (C_2_H_4_O-C_2_H_4_)_x_, comprising 48 mol/L ethylene and 52 mol/L vinyl alcohol (Figure 1 and Figure 2). This polymer is dissolved in dimethyl sulfoxide (DMSO) [22,33].

### 2.2. Granulomeria Protocol

Measurements of particle size distribution were carried out on the device SALD-2201 (Laser Diffraction Particle Size Analyzer, Shimadzu Corporation, Tokyo, Japan). Device control and data processing were performed using WingSALD_II software, Version 2.1.0 (Shimadzu, Tokyo, Japan). Dimethyl sulfoxide (DMSO, Dimexide, 99%, JSC “Tatchempharmpreparaty”, Kazan, Russia) and a storage cuvette with mechanical stirring were used as a dispersion method. The cuvette was thoroughly washed, was filled with DMSO, placed in the device, and the idle signal was measured. The value of the idle signal was automatically entered into the software. The samples were dark opaque liquids in glass vials sealed with polymer septa. The septa were pierced with a sterile 2 mL medical syringe, the needle was washed twice with a sample, then an aliquot of 0.5 mL sample was taken. The sample was injected drop by drop into a storage cuvette with DMSO under constant mechanical stirring. The sample was added before the absorption index fell into the measurement range (determined automatically by the program). Then the measurement was carried out and, based on the results obtained, a report was generated in pdf and xls format.

### 2.3. Microscopy Protocol

Precipitation of particles from a highly dispersed solution onto a silicon substrate:(a)Preparation of highly dispersed nanoparticle solution

The initial suspension was mixed with a solvent in a ratio of 1:5 (suspension:DMSO) to produce a solution. The component was withdrawn using a disposable syringe (suspension) and microdosing device (DMSO). Mixing was performed in a disposable container (Eppendorf), and the total volume of the resulting solution was approximately 1 mL.

The solvent was chosen based on the composition of the nanoparticles studied and the need for its complete evaporation during drying.

(b)Dispersing the solution

The solution was dispersed in an ultrasonic bath for 3–5 min until a homogeneous composition was obtained.

(c)Particle precipitation

For scanning electron microscopy (SEM) studies, the particles were deposited on a clean polished silicon substrate. Precipitation of the prepared solution was carried out using a microdoser. The drying process (evaporation) was conducted in an oven within a fume cupboard.Precipitation of particles from the initial suspension onto a silicon substrate: The initial suspension was precipitated using a disposable syringe. The drying process (evaporation) was conducted on a slab within a fume cupboard. The studies were conducted primarily along the periphery of the precipitated drop.

Microscopy was performed on a Zeiss Auriga Laser workstation (A Carl Zeiss SMT AG Company, Oberkochen, Germany) with intersecting ion and electron beams, additionally equipped with a solid-state laser. The device is equipped with a GEMNI electronic optics column and an auto-emission cathode is used as an electron source. An ion column with a liquid metal ion source forms a focused beam of Ga+ ions with an energy of up to 30 keV.

#### 2.3.1. Scanning Electron Microscopy (SEM)

SEM is designed primarily for obtaining enlarged images of objects up to subnanometer sizes. As the name implies, the image of the objects under study in SEM is formed as a result of scanning a sample with a focused electron beam (a beam of primary electrons) sequentially point by point. At the same time, when the electron beam interacts with the material/surface of the object under study, a large number of different signals are excited. By detecting any of the excited signals, it is possible to build a map of the intensity distribution (microgram) of this signal. The results obtained by using different detectors make it possible to conduct a comprehensive study and form an idea of the object under study.

#### 2.3.2. Energy-Dispersive X-Ray Spectroscopy (EDX)

A focused beam of electrons hitting the sample causes X-ray fluorescence containing both the braking component and the characteristic lines of the atoms of the elements that make up the sample material. By analyzing the X-ray spectrum, it is possible to assess the composition of the sample both qualitatively and quantitatively. A semiconductor detector, cooled to a temperature of approximately −40 °C, was employed for the purpose of recording X-rays. The energy resolution of such a detector is about 120 eV, which does not allow analyzing of energy shifts of characteristic lines due to chemical bonds and also limits the lower threshold of analyzed elements to boron (Z = 5). Elements with lower atomic numbers cannot be reliably detected by the EDX method. The sensitivity of the method depends on the material of the investigated sample and is about 0.1% by weight.

### 2.4. Study Design

This was a non-randomized case-control study. A series of cases was designed to compare the behavior of NAGLEMs containing different tantalum granules during MMA embolization in patients with CSDHs. The study was conducted at the Stavropol Regional Hospital and the Almazov National Research Medical Center, St. Petersburg, Russia. Patients were included in the study from November 2015 to May 2023.

### 2.5. MMA Embolization Technique by NAGLEMs

A 6-7F guiding catheter was placed in the external carotid artery on the side of the hematoma, and a diagnostic double-projection DSA was performed. The anterior and posterior branches of the MMA were identified angiographically (frontal and parietal). Under road-map control, the MMA was catheterized with different types of microcatheters: Headway (17, 21) (MicroVention). MMA microangiography typically (normally) revealed a hypervascular network with a typical “cotton” pattern. The technical details of the procedure were as follows:-Distal catheterization of the frontal and parietal branches of the middle meningeal artery was performed almost to the level where the outer diameter of the microcatheter coincided with the internal lumen of the artery, which is called catheter wedged position.-In a number of cases [38,39,40,41,42], the tortuosity of the MMA (especially at levels where it often passes through the bony canal as a rule [38,39,40,41,42]) did not allow the microcatheter to be guided distally. Consequently, a wedged position could not be achieved. In such instances, we employed the use of arterial spasm that emerged during the course of catheterization attempts. The phenomenon of arterial spasm, occurring in proximity to the microcatheter, effectively impeded the reflux of both the contrast agent and the embolic agent, even when the diameter of the artery exceeded that of the outer diameter of the microcatheter tip. Only after ascertaining the absence of reflux at the DSA did we proceed with the injection of the embolic agent.-Selective digital subtraction angiography (DSA) with the injection of a contrast agent through the microcatheter in the MMA reveals the presence of “cotton” areas of neovascularisation, as well as a network of distal anastomoses and the absence of proximal reflux. From this position, the first portion of NAGLEMs was injected to maximize penetration and obliteration of all MMA branches distal to the tip of the catheter, including newly formed vessels of the CSH capsule, as well as the associated collaterals to the contralateral side, falx anteriorly and posteriorly.-The injection rate of the syringe piston pressure was independent of the type of embolic agent and comparable in the SQUID and ONYX groups.

Embolization should be limited to the distal portions of the frontal and parietal branches of the MMA, preserving the areas of anastomosis with the ophthalmic artery and the petrosal branches of the foraminal segment.

### 2.6. Analysis of NAGLEMs Behavior During MMA Embolization

For convenience in delineating the gel-like composition penetration level, the Frankfort horizontal plane was used as a basis. The auriculo-orbitalis line passes through this plane in the lateral projection. The second line supra-orbitalis as in the scheme of R.U. Krönlein [43] runs parallel to the auriculo-orbitalis line. The third line was added by us to emphasize the most distal blood supply zone of the ipsilateral MMA; it also runs parallel to the first two lines and through the craniometric Median point. Median point is known as the greatest elevation located between Nasion and Bregma on the frontal bone. Nasion isthe point of intersection of the nasofrontal sutures in the median plane. Bregma is the point where the coronal and sagittal sutures meet. The sagittal plane is the main boundary of the vascularization level of the ipsilateral MMA.

Thus, the gel-like composition penetration levels of the ipsilateral MMA can be distinguished as follows:(1)Level I is proximal, bounded by the auriculo-orbitalis and supra-orbitalis lines. This level is typically characterized by the origin of the MMA from the maxillary artery and the presence of a large MMA trunk. In accordance with the Adachi classification modified by Giuffrida-Ruggeri [44,45,46,47,48] the principal MMA trunk may be subdivided into a frontal and a parietal branch in this region, a characteristic feature of MMA types 2b, 3, and 4. Also at this level, the MMA may have the most dangerous anastomoses with the ophthalmic artery and the tympanic branch of the MMA [2,38,40,49].(2)Level II is lateral, bounded by the supra-orbitalis and a line through the craniometric Median point. If the MMA is of type 1 or 2a (according to Adachi as modified by Giuffrida-Ruggeri [44,45,46,47,48] ), the division of the main trunk of the MMA into frontal and parietal branches occurs at this level [44,45,46,47,48]. In this region, second- and third-order branches connecting the frontal and parietal branches of the MMA can frequently be observed [2]. Additionally, in this region, the parietal branch often connects to the posterior meningeal artery, which represents a significant anastomosis with the vertebral artery system.(3)Level III is parasagittal, limited by the line through the craniometric Median point and sagittal plane. At this level, branches are connected with the arteries of the walls of the superior sagittal sinus. There are also direct anastomoses with the anterior circumflex artery and, respectively, with the ophthalmic artery [2].(4)Level IV is contralateral for its ipsilateral MMA. Level IV is characterized by collateral connections with the contralateral meningeal arteries and Falx vessels.

The evaluation scheme according to the outlined protocol is shown in Figure 3. The penetration of the embolic agent into the dural veins, as well as the time and the occurrence of extravasation of the embolic agent during embolization, were considered separately. The results of the evaluation can be found in Section 3.3.

### 2.7. Statistical Analysis

Given the presence in our database of eight patients with CSDHs in whom ONYX 18 (Medtronic) was used for MMA embolization, we selected a comparison group from the database of patients in whom SQUID 18 (Balt) was used (n = 102). To account for potential differences in the patient treatment groups, a pseudorandomization process (PSP) was employed. The groups were balanced through the use of nearest neighbor matching. Matching covariates include demographics (age, gender) and others (e.g., etiology of CSDH formation, baseline mRS severity, transverse dislocation, localization, and thickness of CSDH). Baseline characteristics of groups after PSP were compared. Statistical analysis was performed using StatTech v. 4.4.1 (StatTech LLC, Kazan, Republic of Tatarstan, Russia) and Jamovi 2.5.5 (The jamovi project (2024). jamovi (Version 2.5) [Computer Software]. Retrieved from https://www.jamovi.org). Quantitative variables were assessed for normality using the Shapiro–Wilk test. Quantitative variables following a normal distribution were described using mean (M) and standard deviation (SD); 95% confidence interval (95% CI) for the mean were estimated. Quantitative variables following non normal distribution were described using median (Me) and lower and upper quartiles (Q1–Q3). Categorical data were described with absolute and relative frequencies. Then 95% confidence intervals for proportions were calculated using the Clopper–Pearson method. Comparison of the two groups for a quantitative variable following a normal distribution was performed using Student’s t-test if the variances were equal and Welch’s t-test in the case of unequal variances. The Mann–Whitney U-test was used to compare two groups on a quantitative variable whose distribution differed from the normal distribution. Comparison of frequencies in the analysis of 2 by 2 contingency tables was performed using Fisher’s exact test (for expected values less than 10). When comparing relative rates, we used an odds ratio with corresponding 95% confidence interval (OR 95% CI) as a measure of effect size. In the case of zero values in the cells of the contingency table, the calculation of the odds ratio was performed with the Haldane–Anscombe correction. Comparison of frequencies in the analysis of multifield contingency tables was performed using Pearson’s chi-square test. Differences were considered statistically significant at *p* < 0.05.

## 3. Results

In order to respond to these points, an analysis was conducted on granules of tantalum powder, which constitutes a component of the most commonly utilized NAGLEMs for the embolization of MMA andspecifically for ONYX 18 (Medtronic) and SQUID 18 (Balt). A comparative analysis of the behavior of these gel-like embolic agents was performed on 16 patients with CSDHs.

### 3.1. Granulometry

The granulometry of the sample (ONYX 18) exhibits a considerable range in granule size, with a minimum of 0.255 microns and a maximum of 71.548 microns (Figure 4).

The granulometry of the sample (SQUID 18) reveals a variation in granule size, with a minimum of 0.309 microns and a maximum of 0.818 microns (Figure 5).

The median granule size in sample SQUID 18 is 0.443 ± 0.086 µm (M ± SD), which is an order of magnitude smaller than that observed in the tantalum sample designated as ONYX 18, which is 5.2 ± 0.33 µm (M ± SD). (Figure 6).

### 3.2. Microscopy and Energy-Dispersive X-ray Spectroscopy (EDX)

The second stage of the analysis involved an examination of the shape and dimensions of the tantalum granules in gel-like compositions using electronic microscopy. The sample designated ONYX 18 demonstrated that the granules exhibited an irregular angular shape, a non-porous surface, and a relatively wide dispersion of granules in size. In contrast, the sample SQUID 18 was composed of tantalum granules with a regular spherical shape, a non-porous surface, and a relatively narrow size distribution (Figure 7).

Energy-dispersive X-ray spectroscopy (EDX) has confirmed the presence of tantalum in the studied samples. The amount of tantalum is found to be in close agreement with its atomic content in similar compounds. It is likely that the observed differences in peak intensity are related to the varying shapes of the particle surfaces [50] (Table 2).

### 3.3. Patient Population and Analysis of NAGLEMs Behavior During MMA Embolization

In our clinics ( V.A. Almazov National Research Medical Center in St. Petersburg and Stavropol Regional Hospital, Stavropol, Russia), 16 operations were performed on patients with CSDHs between November 2021 and May 2024. The age of the patients was ranged from 30 to 88 years, with an average age of 62.2 ± 14.3 years (M ± SD). The gender distribution revealed a predominance of male patients (n = 11 (68.8%)). Seven patients (43.8%) presented with bilateral CSDHs, five (31.2%) with a left-sided CSDH, and four (25.0%) with a right-sided CSDH. A recurrence of CSDH was observed in five (31.2%) patients following surgical evacuation of the hematomas. All patients underwent MMA embolization, with bilateral embolization performed in nine (56.2%) cases. In eight (50.0%) cases, ONYX 18 was used as the embolizing agent, while in the remaining eight (50.0%) cases, SQUID 18 was utilized. Following MMA embolization, patients underwent routine brain CT scans at one day, seven days, then at intervals of 30–60 days, 60–90 days, and 90–180 days. This schedule was kept unless clinical circumstances dictated to do otherwise. In case of incomplete resolution of the hematoma after 180 days, further brain CT scans were conducted at 90-day intervals until the hematoma disappeared. The number of days before the hematoma resolution was recorded as a specific number based on the date of the initial CT scan at which it was documented. All patients included in this series demonstrated complete resolution of CSDH, and no additional drainage procedures or repeated embolizations were performed. For further details, please refer to Table 3.

To describe the behavior of NAGLEMs during MMA embolization, the following parameters were evaluated: the level of penetration (Figure 8), microcatheter position, microcatheter diameter, penetration into the dural veins at the first push, assessment of penetration into the dural veins at the end of embolization, the time of embolization, the appearance of extravasation of the gel-like embolic agent, and the time of extravasation from the beginning of embolization (Table 4) (Figure 9).

According to the presented table, when comparing gel-like embolic agents, statistically significant differences were revealed depending on the level of NAGLEMs penetration in dural vessels (*p* = 0.009) (applied method: Pearson’s chi-square test). The diameters of the microcatheters used by both groups were comparable, as well as their positions. No significant relationship was found between these parameters and the degree of penetration.

The injection rate of the syringe piston pressure was independent of the type of embolic agent and was comparable in the SQUID and ONYX groups.

However, this result may be due to the small sample size, and further correction will be made with larger numbers.

### 3.4. Illustrative Cases

In order to illustrate the differences in behavior of the two NAGLEMs during MMA embolization in patients with CSDHs, we present a representative selection of clinical cases.

#### 3.4.1. Case #1

A 51-year-old man suffered a head injury with loss of consciousness 4 months ago and did not seek medical attention. A week before the operation the patient developed motor aphasia; CT scan of the brain revealed a chronic subdural hematoma compressing the left cerebral hemisphere, maximum thickness was 26 mm, and midline shift was 9 mm. MMA embolization was performed using ONYX 18. During embolization it was not possible to achieve distal penetration of the MMA branches on the ipsilateral side, so we had to perform embolization on the contralateral side to ensure penetration of the embolic agent through the midline.

The behavior of the gel-like composition and its penetration level were evaluated according to the above protocol in Section 2.6. (Figure 3).

After embolization, the patient did not require evacuation of the hematoma. Complete resolution of the hematoma was recorded on CT scan of the brain after 215 days.

#### 3.4.2. Case #2

A 63-year-old man with a history of head trauma came to the clinic with complaints of headaches. mRS at the time of admission was 1. CT scan of the brain showed a right-sided chronic subdural hematoma with a maximum thickness of 13 mm and a midline shift of 3 mm. During embolization of the right MMA by ONYX 18, there was extravasation of the gel-like composition on 9.4 min. Extravasation did not lead to clinically significant complications but prevented further embolization (Figure 9).

Resolution of the hematoma was documented by brain CT on day 223 postoperatively. mRs was 0.

#### 3.4.3. Case #3

A 47-year-old woman had a history of a car accident 6 months prior to admission. On admission, she had severe headaches. Occasionally, she had difficulty picking up words. mRs was 2. CT scan of the brain shows a left-sided chronic subdural hematoma with a maximum thickness of 14 mm and a midline displacement of 6.5 mm. During embolization of the left MMA, penetration of the embolizing agent (SQUID 18) into the distal branches of the MMA and its collaterals, as well as penetration into the dural veins at the first push, is noteworthy. At the end of embolization, it was possible to turn off both the arterial flow of the dura mater and its venous outflow (Figure 10).

Complete resolution of the hematoma was recorded on CT scan of the brain 62 days after embolization, mRs was 0.

## 4. Discussion

MMA embolization has proven effective in the treatment of CSDH in randomized case-control studies. CT images of CSDH obtained after embolization of MMA showed contrast enhancement of the dura mater, capsule membrane, septa, and subdural hematomic fluid, which suggested a continuous vascular network between the membranes of CSDH and MMA [2]. This connection is provided by penetrating arterioles with a diameter ranging from 5 to 15 microns. In addition, venous outflow from the dura mater could play an important role in limiting the blood supply to the CSDH capsule. The presence of arteriovenous shunts in dura mater ranging in size from 8 to 12 microns in diameter provides the possibility of embolization of the venous part of dura. Thus, a gel-like composition capable of penetrating into vessels of less than 15 microns is optimal. In 2023, N.R. Ellens et al. [51] conducted a significant meta-analysis comparing various embolic agents for MMA in CSDH. Their findings indicated that the recurrence rate, a necessity for surgical salvage, and periprocedural complications following MMA embolization were not influenced by the specific embolic agent utilized. The principal limitations of this meta-analysis pertain to the retrospective nature of the included studies. These include the considerable variability in embolization techniques. The authors themselves acknowledge that the meta-analysis only included studies that used the same embolic agent in >95% of cases, which may not fully reflect the published experience with MMA embolization. Furthermore, the analysis does not incorporate data on the utilization of SQUID (Balt), which was already documented and disseminated at the time of its publication [4]. Additionally, the findings of randomized trials [15,16,17] on NAGLEMs are not reflected in this meta-analysis. This precludes the formulation of a definitive conclusion, necessitating the conduct of further studies and meta-analyses. Upon evaluation of the shape and granulometry of the two NAGLEMs, a notable discrepancy was observed. The sample SQUID 18 exhibited medium-sized granules with a mean diameter of 0.443 ± 0.086 µm (M ± SD), which were smaller than the tantalum granules in the sample ONYX 18, with a mean diameter of 5.2 ± 0.33 µm (M ± SD). In addition, the tantalum granules in SQUID were of regular spherical shape, with a smaller size spread, and with a smaller surface area of the granules. All this explained the longer sedimentation time of SQUID 18 and the preservation of X-ray contrast estimated in other studies [29,30]. In addition to alterations in the visualization of embolic material throughout the procedure, some authors [25,52] have indicated the potential for premature microcatheter occlusion resulting from the relatively large size of tantalum granules present in ONYX (Medtronic). However, in none of our cases was there a problem with loss of visualization and occlusion of the microcatheter.

It can be observed that the composition of the two most common representatives of NAGLEMs includes tantalum powders that are not only of a different size, but also exhibit a variation in granule shape. It is possible that these characteristics of tantalum particles may provide an explanation for the observation that episodes of extravasation occurred with greater frequency and at a faster rate when ONYX 18 was distributed throughout the branches of MMA. The average time for these episodes was 5.30 min. In cases of dural artery embolization, this factor did not result in clinically significant complications, given that the rupture occurs within the dura mater. Nevertheless, in all such cases, this resulted in the cessation of further embolization, given that the adequate distribution of NAGLEM was no longer feasible, with the gel-like composition leaking through the ruptured vasculature. The rate of injection under syringe piston pressure was found to be independent of the type of embolic agent and comparable in the SQUID and ONYX groups. However, it should be noted that this is a challenging parameter to monitor and objectify. In future studies, it is essential to employ a standardized approach across all groups to ensure meaningful evaluation, enhance reproducibility of results, and minimize the influence of confounding factors. Nevertheless, in the group undergoing MMA embolization for CSDH, particularly using the wedging catheterization technique, extravasations were observed when the embolic agent was injected, potentially due to the neoangiogenetic vessels of the hematoma capsule [3]. It seems that the dimensions and configuration of the tantalum granules constrained the capacity of ONYX 18 to penetrate the dural veins. In our series, dural embolization was observed exclusively in the group utilizing SQUID 18. As previously discussed [4], the minimal viscosity of SQUID 12 confers an advantage in terms of penetration ability in MMA embolization. However, the shape and size of the tantalum granules may also play a key role in penetration. It should be noted that this study is not without limitations, as histological studies of the embolized dura mater were not conducted. However, the lack of experimental studies using microscopy on live specimens limits the understanding of the extravasation process of venous penetration in dural vessels. Despite the different behavior of the embolic agents during the procedure, we found no significant difference in complication rates or outcomes, nor in the rate of hematoma resolution. Further research will undoubtedly contribute to a more comprehensive understanding of this phenomenon.

In regard to the characteristics of tantalum powder granules, ONYX and SQUID represent two distinct embolic agents. In addition to radiopacity, the characteristics of tantalum may also influence the behavior of these embolic agents, including penetration, frequency, and timing of extravasation and the ability to provide more stable occlusion of high-flow fistulas. The capacity to consider not only the viscosity of the polymer but also the characteristics of the powder enables a novel approach to the selection of an embolic agent. Due to the smaller size of tantalum granules, SQUID embolic agent is more homogeneous, has a smaller volume fraction of tantalum in the gel-like composition, and is less susceptible to sedimentation, which significantly increases its ability to penetrate into the vessels of the microcirculatory bed. Concurrently, the larger polymorphic tantalum granules observed in the ONYX 18 sample may account for the stability and predictability of this composition when embolizing larger vessels.

## 5. Conclusions

The use of NAGLEMS with identical viscosity and varying tantalum powders demonstrated disparate levels of penetration into dural vessels. Accordingly, not only the viscosity, but also the size and shape of the tantalum particles may also influence the penetration ability of these gel compositions.

The availability of a range of gel-like embolic agents with varying degrees of penetration enables the endovascular surgeon to select the most appropriate solution for each specific task.

## Figures and Tables

**Figure 1 jfb-15-00319-f001:**
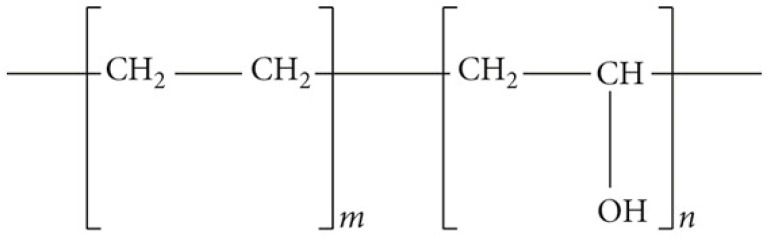
Schematic chemical formula of structure of ethylene vinyl alcohol copolymer (EVOH) (C_2_H_4_O-C_2_H_4_)_x_ [37].

**Figure 2 jfb-15-00319-f002:**
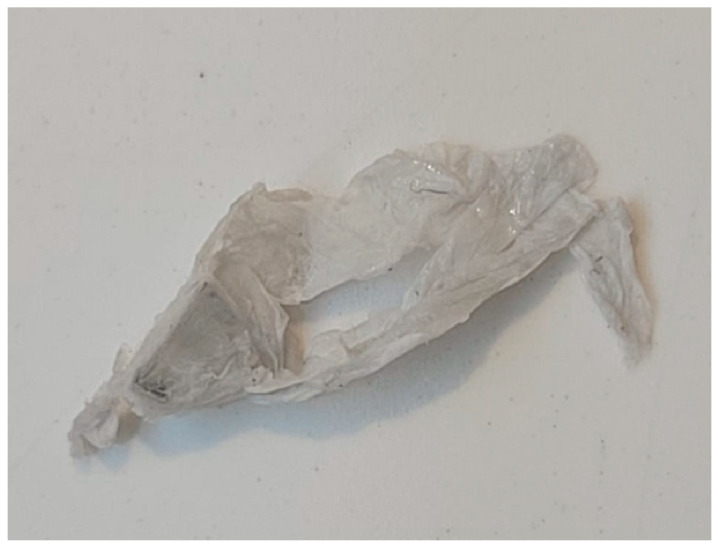
Photograph of the polymer (C_2_H_4_O-C_2_H_4_)_x_ appearance after extraction of tantalum powder and DMSO elimination from Sample 1.

**Figure 3 jfb-15-00319-f003:**
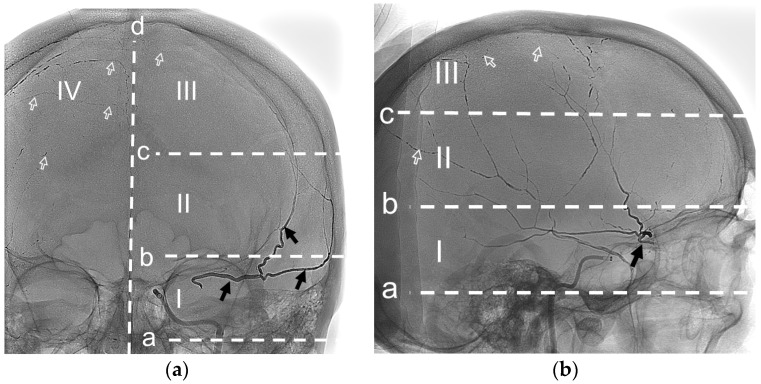
Posterior–anterior (**a**) and lateral (**b**) radiographic images show the radiopaque cast in both MMAs (arrows). The black arrows- ipsilateral MMA (**left side**) with level II penetration by ONYX 18. The white frame arrows—contralateral MMA (**right side**) with level III penetration by ONYX 18. a—auriculo-orbitalis line; b—supra-orbitalis line; c—median point line. I—proximal, II—lateral, III—parasagittal, IV—contralateral levels of penetration (Case #1, Refer to Section 3.4.1).

**Figure 4 jfb-15-00319-f004:**
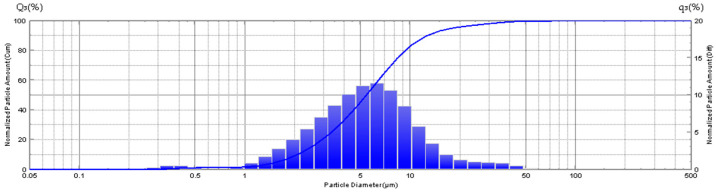
Diagram showing the distribution of tantalum particles in the gel-like composition—ONYX 18.

**Figure 5 jfb-15-00319-f005:**
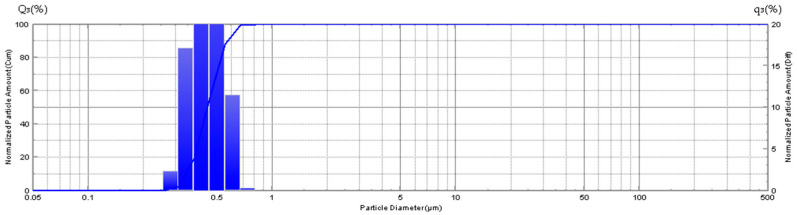
Diagram showing the size (in µm) distribution of tantalum particles in t gel-like composition—SQUID 18.

**Figure 6 jfb-15-00319-f006:**
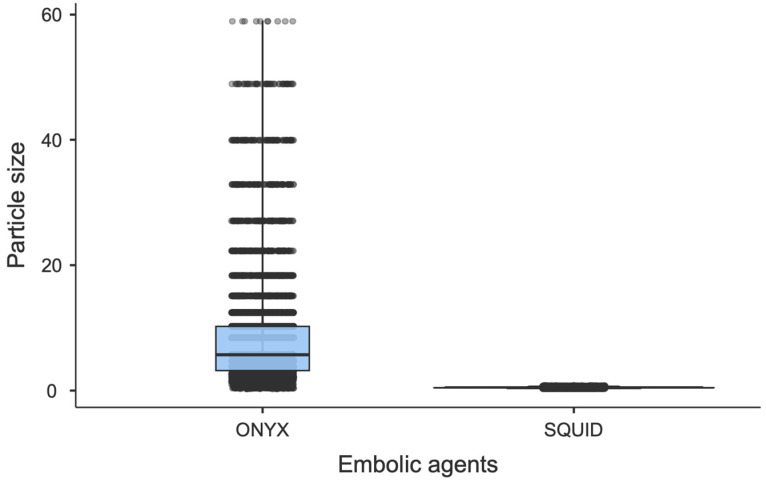
Diagram showing comparison of tantalum particle size (in µm) distributions in gel-like compositions (ONYX 18 and SQUID 18 samples).

**Figure 7 jfb-15-00319-f007:**
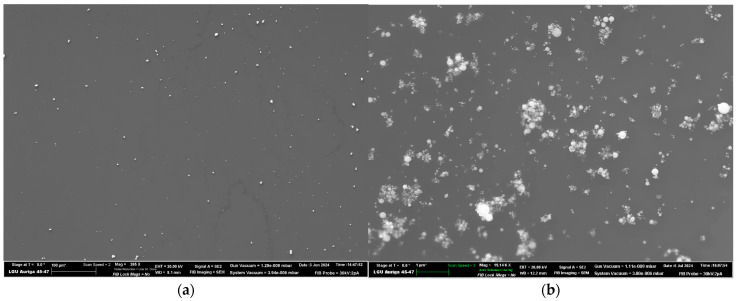
Microscopic images and energy-dispersive X-ray spectroscopy (EDX) of tantalum powder granules following the dispersion of a solution of non-adhesive gel-like embolic materials and subsequent deposition on a silicon substrate. (**a**,**c**) tantalum powder of ONYX 18 and traces of EVOH solution post-dispersion, (**b**,**d**) tantalum powder of SQUID 18 and traces of EVOH solution post-dispersion. Energy-dispersive X-ray spectroscopy (EDX) is presented in (**e**,**g**) spectrum 1 for tantalum granules in ONYX 18 and in (**f**,**h**) spectrum 2 for tantalum granules in SQUID 18.

**Figure 8 jfb-15-00319-f008:**
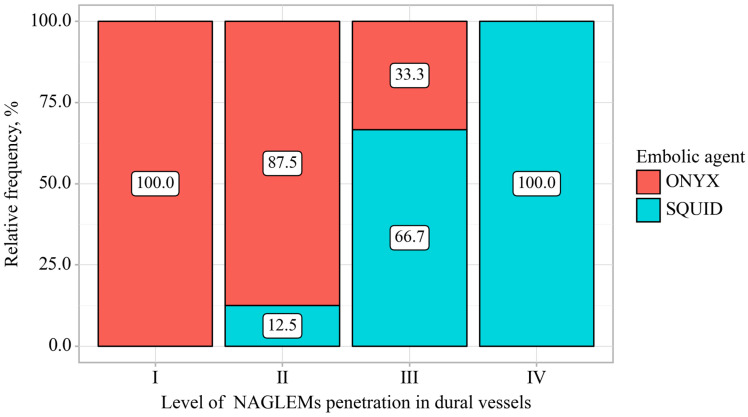
Analysis of gels conditioning on level of NAGLEMs penetration in dural vessels.

**Figure 9 jfb-15-00319-f009:**
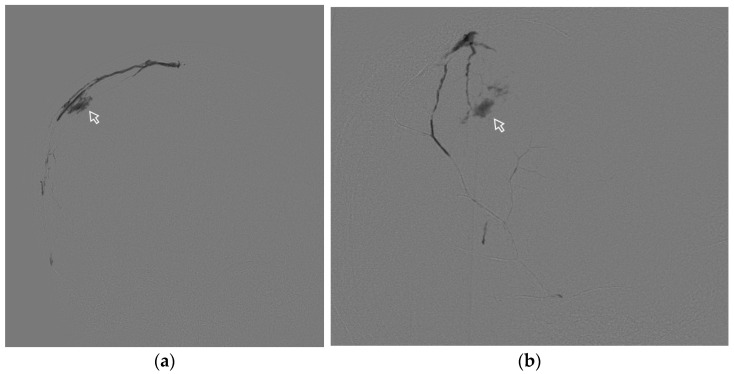
Posterior-anterior (**a**) and lateral (**b**) DSA images show the process of spreading NAGLEM (ONYX 18) along the branches of the right MMA. The white frame arrows indicate extravasation of the embolic agent (ONYX 18).

**Figure 10 jfb-15-00319-f010:**
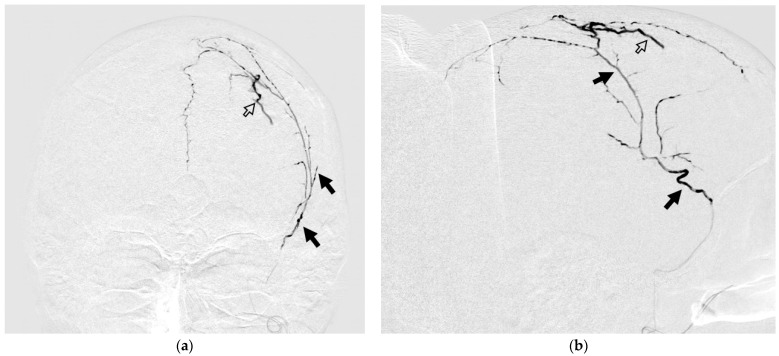
Posterior-anterior (**a**) and lateral (**b**) DSA images show the process of spreading NAGLEM (SQUID 18) along the branches of the left MMA. The black arrows indicate the branches of the left MMA. The black framed arrow indicates the penetration of the embolic agent during the first push. Posterior–anterior (**c**) and lateral (**d**) radiographic images show the radiopaque cast (SQUID 18) in left MMA and dural veins.

**Table 1 jfb-15-00319-t001:** Main characteristics of the most used NAGLEMs (ONYX and SQUID) according to the literature data.

Embolic Agent	Polymer	Polymer Concentration (%)	Viscosity at 40 °C, cSt	Contrast Agent	Tantalum Concentration (%)	Tantalum Granule Size (µm)	Minimal Vessel Diameter (µm)
Onyx 18	EVOH (EVAL) [19,31,32,33]	6.0 [19,25]	18 [19,25]	Tantalum grain [31]	35 [34]	0.7–22 [35]	150 [36]
Onyx 20		6.5 [19,25]	20 [19,25]		35 [34]	0.7–22 [35]	150 [36]
Onyx 34		8.0 [19,25]	34 [19,25]		35 [34]	0.7–22 [35]	150 [36]
Squid 12	EVOH (EVASIN) [19,31,32,34]	4.0 [25,34]	12 [25,34]	“micronized” tantalum grain [34]	30 [34]	<1 [34]	no data
Squid 12 LD		4.0 [25,34]	12 [25,34]		20 [34]	<1 [34]	no data
Squid 18		5.3 [25,34]	18 [25,34]		30 [34]	<1 [34]	no data
Squid 18 LD		5.3 [25,34]	18 [25,34]		20 [34]	<1 [34]	no data

**Table 2 jfb-15-00319-t002:** Summary of energy dispersive X-ray spectroscopy (EDX) for tantalum powder in ONYX (spectrum 1) and SQUID (spectrum 2).

Element	Line Type	Weight %	Sigma Weight %	Atom. %
Spectrum 1 (tantalum powder in ONYX 18)				
Ta	L-series	100.0	0.00	100.0
Spectrum 2 (tantalum powder in SQUID 18)				
C	K-series	26.01	2.54	70.52
O	K-series	8.71	1.41	17.72
Ta	L-series	65.29	2.61	11.75

**Table 3 jfb-15-00319-t003:** Summary of the demographic and treatment data of two patient groups treated by NAGLEM embolization.

Variables	NAGLEMs		*p*
	ONYX 18 (n = 8)	SQUID 18 (n = 8)	
Patient age (years), M (SD)	64.75 (12.98)	59.63 (15.89)	0.492
Sex female/male, abs. (%)	2 (25.0%)/6 (75.0%)	3 (37.5%)/5 (62.5%)	1.000
CSDH location, abs. (%)			
Bilateral	3 (37.5%)	4 (50.0%)	0.842
Left unilateral	3 (37.5%)	2 (25.0%)	
Right unilateral	2 (25.0%)	2 (25.0%)	
Previous surgery evacuation	3 (37.5%)	2 (25.0%)	1.000
Causative factor, abs. (%)			
anticoagulant therapy	2 (25.0%)	1 (12.5%)	1.000
trauma	6 (75.0%)	7 (87.5%)	
mRS at the time of admission, M (SD)	1.38 (1.19)	1.25 (0.71)	0.802
Midline shift at the time of admission (mm), M (SD)	3.50 (3.59)	4.38 (4.06)	0.655
Maximum thickness of the left ChSDH (mm), M (SD)	15.17 (7.99)	18.08 (6.67)	0.508
Maximum thickness of the right ChSDH (mm), M (SD)	16.43 (8.46)	18.90 (7.11)	0.565
Side of MMA embolization, abs. (%)			
Bilateral	5 (62.5%)	4 (50.0%)	0.801
Left side	1 (12.5%)	2 (25.0%)	
Right side	2 (25.0%)	2 (25.0%)	
Days of total ChSDH resolution (days), Me [IQR]	195.00 [183.75; 217.00]	191.00 [154.25; 199.50]	0.343

**Table 4 jfb-15-00319-t004:** The behavior of NAGLEM during MMA embolization was observed and documented (bilateral embolizations were taken into account, as detailed in Table 3).

Variables	NAGLEMs		*p*
	ONYX 18 (n = 8)	SQUID 18 (n = 8)	
Penetration levels NAGLEMs			
I	1 (9.1%)	0 (0.0%)	0.009
II	7 (63.6%)	1 (8.3%)	
III	3 (27.3%)	6 (50.0%)	
IV	0 (0.0%)	5 (41.7%)	
Level of microcatheter position in MMA			
I	3 (27.3%)	1 (8.3%)	0.462
II	7 (63.6%)	9 (75.0%)	
III	1 (9.1%)	2 (16.7%)	
Microcatheter, abs. (%)			
Headway 17	9 (81.8%)	9 (75.0%)	1.000
Headway 21	2 (18.2%)	3 (25.0%)	
Penetration into the dural veins at the first push	0 (0.0%)	3 (37.5%)	0.200
Penetration into the dural veins at the end of embolization	3 (37.5%)	8 (100.0%)	0.026 *
NAGLEMs extravasation	7 (87.5%)	2 (25.0%)	0.041 *
Average time of the appearance of extravasation from the beginning of embolization NAGLEMs (min) Me [IQR]	5.30 [5.00; 6.25]	11.20 [9.20; 12.30]	0.012 *

*—Differences are statistically significant (*p* < 0.05).

## Data Availability

The original contributions presented in the study are included in the article, further inquiries can be directed to the corresponding author. The data are not publicly available due to the privacy of the patients who assisted in the research.

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
