# Peer review of "Penetration of Non-Adhesive Gel-like Embolic Materials During Dural Vessels Embolization According to Characteristics of Tantalum Powder"

_jfb, 2024, doi:10.3390/jfb15110319_

Round 1

Reviewer 1 Report

Comments and Suggestions for Authors

Thank you for the opportunity to review this paper on the influence of particle size and shape in NAGLEMs regarding MMA embolization. It sheds light on an aspect that has received little attention to date and is highly relevant for the increasing number of endovascular treatments of SDH.

The wealth of analyses carried out is considerable. Nevertheless I have a few points to make, the last is of highest imprtance.

- Is there a supplement or an appendix? If not, please delete the comments on page 15.

- page 6 line 202-208: Please refer to Figure 8 to enhance the understanding of how the authors made their assessment of the penetration depth. While reading the protocol I wished to see a drawing and only later I saw the case-example with it!

- page 6, line 209: mismatch of text and figures "The evaluation scheme according to the outlined protocol is shown in Figure 5. The results of the evaluation can be found in Table 2." Isn`t it Figure 7 and Table 4?

- Table 3 only provides data  for"days of total resolution" - Have all SDH healed completely without further surgery or second embolization? With what modality and at what intervals was this checked?

- What about the other NAGLEMs available on the market? PHIL, Menox, LAVA? The reader would benefit greatly from a brief assessment of their particle sizes and shape

- The high rate of extravasation in both groups was surprising. Likewise, the statement that in the absence of distal penetration, an additional contralateral approach was used (page 12 line 323 ff.). Is there any literature on the question of whether such a forced procedure or occlusion of veins is even necessary? This contradicts the theory that neither type nor localization of the embolization material (liquid or even coils) have an influence on the recurrence rate. https://doi.org/10.7461/jcen.2023.e2023.04.002  This contradiction must be included in the discussion!

Author Response

Dear Reviewer,

Thank you for your detailed and thoughtful review of our paper on the influence of particle size and shape in NAGLEMs regarding MMA embolization. We appreciate your insights and acknowledge the importance of the points you've raised. Below we address each of your comments:

  • Supplement or Appendix: We apologize for the oversight regarding the comments on page 15. We have removed the relevant comments to provide clarity.
  • Reference to Figure 8 (Page 6, lines 202-208): We thank you for that suggestion. We have revised the text and referenced Figure 8 earlier (now Figure 3) to help understand the depth of penetration assessment. All figures have been renumbered accordingly. The figure has also been included in the protocol section to provide a clearer understanding. At the end of the description of figure 3, a link to the section where the case is described has been inserted.
  • Mismatch of Text and Figures (Page 6, line 209): We appreciate your keen eye for detail. Since we have rearranged the figures, this is now figure 3. We replaced the reference to the table (because the rules require the table to be cited immediately after the first mention in the text) with a reference to section 3.3. Thank you for your valuable comment, we hope that our corrections have made the text easier to understand.
  • Table 3 and SDH Resolution: We are pleased to confirm that complete resolution of CSDH was noted in all cases, without the need for additional surgery or re-embolisation. We have included detailed information on the methods and intervals used for validation, and have incorporated the data into the revised manuscript.
  • Comparison with Other NAGLEMs: We appreciate the interest expressed by readers in other NAGLEMs such as PHIL, Menox, and LAVA and agree that they are worthy of further discussion. However, we have chosen to focus our attention on other topics for the time being. We have selected the most commonly used embolytes for chronic subdural haematomas, SQUID and ONYX, as there is already a wealth of clinical data available on them. It is possible that the size of tantalum granules in the dura mater vessels may make a difference. PHIL does not contain tantalum granules in its composition; iodine was used as a contrast agent. LAVA is primarily an embolic agent for peripheral vessels, and there is limited data on its behaviour in dura mater vessels for Menox. We believe that this issue may warrant further consideration, potentially within the context of a review article. In this paper, we are concerned that it may prove confusing for the reader.
  • High Rate of Extravasation and Contralateral Approach (Page 12, line 323 ff.): We are grateful for your consideration of this important matter. We have taken the liberty of including a comprehensive paragraph and providing additional references to the discussion.

We are committed to improving our manuscript in line with your recommendations and hope these changes will meet your expectations. Thank you once again for your invaluable feedback.

Best regards,
The Authors

Reviewer 2 Report

Comments and Suggestions for Authors

An interesting work in which the authors undertook to evaluate the difference in the size of tantalum granules used in two different embolization agents from the group of Non-Adhesive Gel-like Embolic Materials (NAGLEMs), which are widely used for embolization of branches of the middle meningeal artery (MMA) in patients with chronic subdural hematomas.

In the work, the authors discussed in detail the significance of the size of the embolization agent particles on the ability to penetrate the so-called penetrating arterioles, which provide a connection between the inner surface membrane and the membrane of chronic subdural hematoma. Their diameter ranges from 5 to 15 micrometers.

To analyze the granules of tantalum powder, which are part of the most popular NAGLEMs, two samples were selected from a batch used in standard clinical practice: sample 1 – gel-like composition ONYX 18 1.5 ml and sample 2 – gel-like composition SQUID 18 1.5 ml. The authors initially assumed that not only the viscosity of NAGLEM, but also the size and shape of tantalum granules may be important for the penetration of these substances. In the conducted analysis, they noticed that SQUID 18 samples contain granules that are ten times smaller than tantalum granules in the ONYX 18 sample - 0.443±0.086 (M±SD) vs. 5.2±0.33 (M±SD). Tantalum granules in SQUID 18 have a regular spherical shape, in ONYX 18 they have an irregular angular shape.

Comparing the behavior of NAGLEM of the same viscosity during MMA embolization in patients with CSDH (mean age 62.2 ± 14.3 years), in the group in which SQUID 18 was used (n=8), the drug with smaller granules showed significantly greater penetration capacity in the vessels of the dura mater, compared to the group in which ONYX 18 was used (n=8).

In the conclusion, the authors stated that not only the viscosity of NAGLEMs, but also the size and shape of the tantalum granules may have a significant effect on the penetration capacity of NAGLEM.

The work is interesting, the method of analysis was well documented, described in detail.

The literature is extensive and refers in detail to the analysis, discussions and conclusions presented by the authors.

The work also has a practical dimension, showing a large difference between two drugs, which are practically presented on the market as embolization agents and with similar pharmacokinetic parameters and similar indications.

Author Response

Dear Reviewer,

Thank you for your comprehensive and thoughtful review of our manuscript. We are pleased to hear that you found the research interesting and appreciated the detailed documentation of our analysis methods. Your recognition of the practical implications of our findings, particularly the significant differences between the two embolic agents, is greatly appreciated.

Your feedback regarding the significance of particle size and shape on the penetration capacity of NAGLEMs strengthens the importance of our work. We aimed to highlight not only the role of viscosity but also the physical characteristics of tantalum granules, and we are glad to see that this aspect was well received.

We appreciate your acknowledgment of the extensive literature review and how it supports our conclusions and discussions. Your positive remarks encourage us to continue exploring this field and further expand our understanding of embolic agents.

Thank you once again for your valuable feedback and for recognizing the contributions our study makes to clinical practice and research.

Best regards,

Author's team

Reviewer 3 Report

Comments and Suggestions for Authors

Authors conduct a study comparing the tantalum particle size/shape and distal penetration between Squid and Onyx embolic agents.  

Authors mention that the Headway was navigated distally till it was wedged. How does a catheter with about 600-micron outer diameter get wedged in the proximal end of Level 1 (e.g. Case #1)? What are the Levels with respect to the MMA and what is the proximal to distal range of vessel diameters for each Level? The microcatheter tip position should have been controlled between the Onyx and Squid groups and it seems that this was not done. This confounds the clinical results (Table 4, Figure 7) of “significantly greater penetration ability” (abstract line 28)

 Also, was the Headway 17 or 21 use consistent between both groups? I would agree that Squid with lower tantalum granule size would potentially result in better distal penetration, but the clinical results shown seem to exaggerate the difference. Onyx-18 is frequently used to embolize Level III vessels and also contralateral vessels as opposed to their dataset. The difference shown is also likely because of the very low sample size. This needs to be discussed further in the manuscript.  

It was interesting to note that literature data exists to characterize Onyx and Squid as reported by the authors in Table 1. Unfortunately, none of the relevant references in Table 1 are correct. Ref 35 has no mention of 35% tantalum concentration or 150-micron minimum vessel diameter. Ref 36 is an online link that is inaccessible. Ref 37 is about using EVOH as a barrier to migration of substances from paperboard and has nothing to do with Squid or Onyx or tantalum. Other than the well-known viscosity values (part of the product names) and EVOH polymer concentration, none of the other data listed in Table 1 are supported. Also, Ref 38 (Figure 1) is wrong. Ref 39 (line 286) is wrong. Ref 41 (line 382) is wrong and does not mention anything about tantalum sludge formation. Ref 42 (line 382) does not exist. Citations seem to be extremely poor throughout the entire manuscript, which is unacceptable.

The granulometry and microscopy results (Figures 3-6) are certainly valuable. There is no mention of how the samples were prepared for the microscopy protocol (line 135).

The extravasation aspect needs to be explained better. There is no mention of extravasation in the Methods. Was the injection rate consistent between the Squid and Onyx groups, as this will affect pressures?

There was no difference in hematoma resolution outcomes between the two groups – discussion needs to be added.

Lines 221-225: The logistic regression analysis must be corrected. Type of embolic material is a dependent variable and a co-dependent variable (in a binary logistic regression) and a covariate?

Figures 3,4,6 are crucial results, but axes titles/legends/labels are blurred and nearly impossible to read.

Line 209: Figure 5 and Table 2 are not related to evaluation scheme.

Line 40 – Balt clinical trial is STEM not STEP

Author Response

We appreciate your thorough review and valuable feedback on our study comparing tantalum particle size and shape in Squid and Onyx embolic agents.

  1. Microcatheter Wedge Positioning:

Authors mention that the Headway was navigated distally till it was wedged. How does a catheter with about 600-micron outer diameter get wedged in the proximal end of Level 1 (e.g. Case #1)?

In regard to catheter navigation, the objective of our protocol was to place the microcatheter as close as possible to the distal target vessels without causing significant trauma. The second objective was to occlude the artery in order to minimise reflux during embolization injection. Prior to the administration of the embolic agent, the effectiveness of the clamping procedure was confirmed through the injection of contrast medium. Due to the frequent tortuosity of the MMA branches in Level I (where it frequently traverses the bony canal), spasm occurs during the passage of the microcatheter, which contributes to the absence of reflux, even in instances where the diameters are mismatched. We acknowledge that the use of terminology such as "wedge" may be interpreted as occlusion, and we have attempted to provide clarification in the manuscript. In particular, adjustments to the position of the microcatheter tip were made according to the dimensions of the vessel in question. However, it is acknowledged that the implementation of a standardised approach across all groups would enhance the reproducibility of the results and minimise the influence of confounding factors.

What are the Levels with respect to the MMA and what is the proximal to distal range of vessel diameters for each Level? 

Thank you for your interest and valuable comments. We regret to inform you that we do not currently have statistical information on the average vessel diameter at each level. However, we believe that within the scope of this article and material, it would not be very reliable. Nevertheless, we will definitely consider analysing all our material within the framework of another paper and article, and we will make measurements. We appreciate your understanding.

The microcatheter tip position should have been controlled between the Onyx and Squid groups and it seems that this was not done. This confounds the clinical results (Table 4, Figure 7) of “significantly greater penetration ability” (abstract line 28)

 Also, was the Headway 17 or 21 use consistent between both groups? 

Thank you for your valuable comment, indeed by the position of the microcatheter and the catheterization technique used, both groups were relatively oospostable. We have added these key points to Table 4.

Furthermore, an additional discussion of the results has been included below the table, and we are in full agreement with the assessment of the limitations of the sample size. The analysis yielded the following results.

  1. Citations and Literature References:

We sincerely apologise for errors in our references. Unfortunately it is really difficult to find information about the characteristics of such semi-popular embolic agents. That is why we decided to use even such sources, which are located on web resources. For convenience of familiarisation with them we have attached them as accompanying materials.

In the source number 35 ‘Onyx Liquid Embolic System (LES) Appendix for the Instruction for Use Available online: https://docs.nevacert.ru/files/med_reestr_v2/39314_instruction.pdf (accessed on 21 August 2024)’.  in the table on page 3 The minimum vessel size, In which embolic liquid must be able to penetrate - 150.

We have also added reference 36 ‘SQUID Liquid Embolic Agent (User Manual) Available online: https://docs.nevacert.ru/files/med_reestr_v2/23334_instruction.pdf (accessed on 21 August 2024)’ to the additional files.  where on page 8 in the table there are data on tantalum powder concentration (the only limitation is that the manual is in Russian).

Ref 41 (line 382) is wrong and does not mention anything about tantalum sludge formation. Ref 42 (line 382) does not exist

Onyx is made radiopaque by the addition of micronized tantalum. Several drawbacks have been observed with this mixture, including the need to homogeneously suspend the tantalum powder prior to use, inhomogeneous opacification due to rather rapid sedimentation of the suspended tantalum during prolonged delivery, and the associated risk of catheter clogging due to inhomogeneity of the material. -
Kulcsár Z, Karol A, Kronen PW, Svende P, Klein K, Jordan O, Wanke I. A novel, non-adhesive, precipitating liquid embolic implant with intrinsic radiopacity: feasibility and safety animal study. Eur Radiol. 2017 Mar;27(3):1248-1256. doi: 10.1007/s00330-016-4463-7. Epub 2016 Jun 14. PMID: 27300197.

However, the manufacturer notes thatthe tantalum particle size in Squidperi is lower thanthat in Onyx™. This provides a lower sedimentation and higher homogeneity of solution, with tantalum particles sedimenting in it twice as slowly. In addition, additional micronization of tantalum excludes microcatheter occlusion. -
Modern liquid embolizing agents based on polymers: composition, properties and applications (review) DOI:10.31044/1994-6260-2021-0-6-3-13

  1. Microscopy Protocol Details:The sample preparation process for microscopy has been added and described in detail to describe the fixation and sectioning steps used prior to analysis.

  1. Extravasation Explanation:We have added discussion of this problem to the appropriate section.
  2. Hematoma Resolution Outcomes:We have added discussion of this problem to the appropriate section.
  3. Logistic Regression Analysis Correction:

We will refrain from writing a logistic regression model so as not to confuse the reader.

  1. Improvement of Figure Legibility:Figures and graphics with legends are hardware graphics, perhaps in the pdf version of the manuscript they do lose clarity, so we have attached them as separate files
  2. Correction of Errors:

 All errors are corrected

Thank you so much for taking the time to read our manuscript, we really appreciate your careful reading and understand how time consuming it is. Thank you again, we hope we have been able to answer your questions.

The Authors

Round 2

Reviewer 1 Report

Comments and Suggestions for Authors

Thank you for the revised version of your work!

Most of the points became much clearer and the additional information led to a better understanding.

I just didn`t understand the new paragraph side 6 line 198-201! Please clarify!

Comments on the Quality of English Language

side 6 line 198-201 not clear

Author Response

Thank you for the positive feedback on our revised manuscript. We appreciate your detailed review and the opportunity to improve our work further. Below, we address your specific concern regarding the paragraph on page 6, lines 198-201:

  • Clarification: In this section we discuss the position of the catheter. This section explains why there are different levels of microcatheter tip position in Table 4 (results). Unfortunately, the wording may have been ambiguous. We have revised this section to ensure clarity and accuracy by clearly stating.

It now reads as follows:

«In a number of cases, the tortuosity of the MMA (especially at levels where it often passes through the bony canal as a rule) did not allow the microcatheter to be guided distally. Consequently, a wedged position could not be achieved. In such instances, we employed the use of arterial spasm that emerged during the course of catheterisation attempts. The phenomenon of arterial spasm, occurring in proximity to the microcatheter, effectively impeded the reflux of both the contrast agent and the embolic agent, even when the diameter of the artery exceeded that of the outer diameter of the microcatheter tip. Only after ascertaining the absence of reflux at the DSA did we proceed with the injection of the embolic agent»

Thank you once again for your constructive feedback.

Reviewer 3 Report

Comments and Suggestions for Authors

Authors have made several changes, most notably on microcatheter position and size comparison as well as correcting citations.

Sample preparation for microscopy needs more clarity on how only the tantalum was extracted from Onyx and Squid. Diluting the samples 1:5 and depositing it on a clean silicon substrate extracts tantalum only? In other words, are the SEM images in Figure 7 of EVOH+tantalum or tantalum only?

Lines 255 to 257 in updated manuscript still describe logistic regression, which is poorly explained.

What are the Levels with respect to the MMA even if diameters cannot be listed. Where is the origin of the MMA or bifurcation of MMA or first bifurcation of anterior or posterior division?

Please add author response to Discussion section: “… the implementation of a standardised approach across all groups would enhance the reproducibility of the results and minimise the influence of confounding factors.”

Please reword line 438 – the meta-analysis only included studies if >95% of cases were done with the same embolic agent. This is not clear in the current wording.

Line 352 – extravasation is Figure 9, not Figure 8.

Author Response

Dear Reviewer,

Thank you for your valuable feedback and for highlighting these important points. We appreciate the opportunity to clarify and improve our manuscript.

Reviewer Comment: Authors have made several changes, most notably on microcatheter position and size comparison as well as correcting citations. Sample preparation for microscopy needs more clarity on how only the tantalum was extracted from Onyx and Squid. Diluting the samples 1:5 and depositing it on a clean silicon substrate extracts tantalum only? In other words, are the SEM images in Figure 7 of EVOH+tantalum or tantalum only?

Response: Thank you for pointing this out. We have updated the manuscript to clarify that during sample preparation, the procedure involved the use of dimethyl sulfoxide (DMSO) to dissolve surrounding materials, allowing tantalum particles to be deposited on the substrate after drying. The SEM images in Figure 7 predominantly show tantalum particles, although traces of EVOH may still be present due to incomplete removal.

Reviewer Comment: Lines 255 to 257 in updated manuscript still describe logistic regression, which is poorly explained.

Response: We apologize for any confusion. We have revised these lines to provide a clearer explanation of the PSP.

Reviewer Comment: What are the Levels with respect to the MMA even if diameters cannot be listed. Where is the origin of the MMA or bifurcation of MMA or first bifurcation of anterior or posterior division?

Response: The manuscript has been updated to provide a more detailed description of the Levels with respect to the middle meningeal artery (MMA). This includes information about the origin, bifurcation, and divisions of the MMA to aid in understanding vessel anatomy in relation to the described Levels.

Reviewer Comment: Please add author response to Discussion section: “… the implementation of a standardised approach across all groups would enhance the reproducibility of the results and minimise the influence of confounding factors.”

Response: The suggested sentence has been incorporated into the Discussion section to emphasize the importance of implementing a standardized approach in future studies to improve result reproducibility and reduce confounding factors.

Reviewer Comment: Please reword line 438 – the meta-analysis only included studies if >95% of cases were done with the same embolic agent. This is not clear in the current wording.

Response: We have rephrased line 438 to clearly state that the meta-analysis included only those studies where more than 95% of the cases used the same embolic agent, ensuring consistency within the analyzed sample.

Reviewer Comment: Line 352 – extravasation is Figure 9, not Figure 8.

Response: Thank you for catching this error. The reference has been corrected to indicate that extravasation is shown in Figure 9, not Figure 8.

We sincerely thank the reviewer for their insightful comments and valuable suggestions, which have significantly contributed to improving the clarity and quality of our manuscript. Your feedback is much appreciated.

Best regards,

Team of authors